# 6′-*O*-Caffeoylarbutin from Quezui Tea: A Highly Effective and Safe Tyrosinase Inhibitor

**DOI:** 10.3390/ijms25020972

**Published:** 2024-01-12

**Authors:** Dong Xie, Wengan Fu, Tiantian Yuan, Kangjia Han, Yuxiu Lv, Qi Wang, Qian Jiang, Yingjun Zhang, Guolei Zhu, Junming Xu, Ping Zhao, Xiaoqin Yang

**Affiliations:** 1Laboratory of National Forestry and Grassland Administration on Highly-Efficient Utilization of Forestry Biomass Resources in Southwest China, Southwest Forestry University, Kunming 650224, China; dong-xie@outlook.com (D.X.); 18751856651@163.com (W.F.); 19387322404@163.com (T.Y.); xqyhankangjia6174@163.com (K.H.); lvyuxiu0130@outlook.com (Y.L.); qiwang202312@163.com (Q.W.); m17368708632@163.com (Q.J.); guoleizhu@163.com (G.Z.); 2Laboratory of Phytochemistry and Plant Resources in West China, Kunming Institute of Botany, Chinese Academy of Sciences, Kunming 650204, China; zhangyj@mail.kib.ac.cn; 3Institute of Chemical Industry of Forest Products, Chinese Academy of Forestry, Nanjing 210042, China; xujunming@icifp.cn

**Keywords:** 6′-*O*-caffeoylarbutin, tyrosinase inhibitor, inhibitory mechanism, computational simulation, food preservation

## Abstract

Tyrosinase is vital in fruit and vegetable browning and melanin synthesis, crucial for food preservation and pharmaceuticals. We investigated 6′-*O*-caffeoylarbutin’s inhibition, safety, and preservation on tyrosinase. Using HPLC, we analyzed its effect on mushroom tyrosinase and confirmed reversible competitive inhibition. UV_vis and fluorescence spectroscopy revealed a stable complex formation with specific binding, causing enzyme conformational changes. Molecular docking and simulations highlighted strong binding, enabled by hydrogen bonds and hydrophobic interactions. Cellular tests showed growth reduction of A375 cells with mild HaCaT cell toxicity, indicating favorable safety. Animal experiments demonstrated slight toxicity within safe doses. Preservation trials on apple juice showcased 6′-*O*-caffeoylarbutin’s potential in reducing browning. In essence, this study reveals intricate mechanisms and applications of 6′-*O*-caffeoylarbutin as an effective tyrosinase inhibitor, emphasizing its importance in food preservation and pharmaceuticals. Our research enhances understanding in this field, laying a solid foundation for future exploration.

## 1. Introduction

Tyrosinase (EC 1.14.18.1), also referred to as polyphenol oxidase, is widely distributed across plants, animals, and microorganisms [1,2]. Illustrated in Appendix A, tyrosinase functions as a pivotal, rate-limiting, multifunctional copper (II)-containing metalloenzyme that plays a crucial role in melanin biosynthesis [3]. The primary biochemical functions of tyrosinase encompass the hydroxylation of L-tyrosine into L-dihydroxyphenylalanine (L-DOPA), followed by the subsequent oxidation of L-DOPA leading to the formation of DOPA-quinone. Additionally, tyrosinase has the potential to facilitate the oxidation of 5,6-dihydroxyindole, resulting in the production of indole-5,6-quinone [4]. In plants, tyrosinase orchestrates the conversion of phenolic compounds into quinone substances. Subsequent polymerization of these quinones generates pigments accountable for the occurrence of browning phenomena [5]. This process directly impacts the inherent visual appeal and nutritional value of various foods like fruits, juices, and vegetables, consequently diminishing their market value. Hence, the inhibition of tyrosinase activity and interception of the DOPA oxidation pathway have emerged as robust strategies in notably curbing melanin production, particularly in contexts that involve counteracting the browning processes.

The utilization of effective and safe tyrosinase inhibitors has the potential to maintain the quality of fruits and vegetables while also extending their shelf life. However, whether derived from natural sources or synthesized through organic compounds, these inhibitors have faced challenges in seamlessly integrating into the food industry due to limitations such as compromised safety, reduced activity, and insufficient solubility [6]. Consequently, the exploration of novel tyrosinase inhibitors continues to be a growing focal point within the modern realm of food research.

Recently, a variety of natural compounds, including phenols, flavonoids, hydroxystilbenes, and cinnamic acid derivatives, have been explored as potential tyrosinase inhibitors due to their ability to reduce tyrosinase activity and suppress pheomelanin production [7]. The increasing attention towards the safety profiles and diverse characteristics of these natural-source-derived tyrosinase inhibitors is noteworthy [8]. The naturally occurring arbutin derivative, 6′-*O*-caffeoylarbutin (Figure 1), is sourced from Quezui Tea, often utilized for medicinal purposes [9,10,11]. A previous phytochemical study revealed an exceptionally high 6′-*O*-caffeoylarbutin content, up to 22%, in Quezui tea processed from the tender leaf buds of *Vaccinium dunalianum Wight* [12]. Subsequent investigations utilizing a live zebrafish model, an excellent platform for probing melanocyte development and pigment disease pathobiology, demonstrated a dose-dependent inhibitory effect of 6′-*O*-caffeoylarbutin on melanin production. This inhibition proved approximately twice as potent as that of beta-arbutin, with toxicity levels around half that of beta-arbutin. Moreover, the removal of 6′-*O*-caffeoylarbutin led to complete recovery of melanin synthesis [13]. These findings highlight the potential of 6′-*O*-caffeoylarbutin as a natural candidate for inhibiting tyrosinase, characterized by its strong inhibition of melanin production.

Nevertheless, the effects of 6′-*O*-caffeoylarbutin on tyrosinase activity and its binding mechanism remain uncertain. Therefore, it is imperative to investigate the mechanism of 6′-*O*-caffeoylarbutin-induced tyrosinase inhibition by comprehensively examining the conformational characteristics of the tyrosinase-6′-*O*-caffeoylarbutin complex at the molecular level. Mushroom tyrosinase (mTyr) derived from Agaricus bisporus stands as a prominent and cost-effective source of active tyrosinase, serving as the exclusive commercially accessible variant [14]. Consequently, our study undertakes a comprehensive exploration to elucidate the potential mechanism of 6′-*O*-caffeoylarbutin-induced mTyr activity inhibition. This investigation encompasses various techniques, including high-performance liquid chromatography (HPLC), ultraviolet_visible (UV_Vis) spectroscopy, fluorescence spectroscopy, and computational simulation, all aimed at understanding the enzyme_inhibitor interactions. Additionally, we undertook a safety assessment of 6′-*O*-caffeoylarbutin and conducted preliminary research on its preservation efficacy in apple juice. The outcomes of this study furnish both a theoretical foundation and a practical basis for the innovation of novel fruit and vegetable preservation methods.

## 2. Results and Discussion

### 2.1. mTyr Inhibition Mechanism of 6′-O-Caffeoylarbutin

#### 2.1.1. Inhibitory Effect of 6′-*O*-Caffeoylarbutin on Oxidation of L-DOPA Catalyzed by mTyr

6′-*O*-caffeoylarbutin has garnered significant attention due to its abundant presence in Quezui tea and its remarkable anti-melanin activity. Previous research has demonstrated that 6′-*O*-caffeoylarbutin exhibits superior anti-melanin activity compared to the commercial whitening agent beta-arbutin, with the effects of melanin production being fully restored upon removal of 6′-*O*-caffeoylarbutin. As shown in Appendix A, structural analysis reveals that 6′-*O*-caffeoylarbutin is derived from the beta-arbutin structure by adding the caffeic acid moiety with two hydroxyl groups. This modification endows 6′-*O*-caffeoylarbutin with both monophenol and diphenol functionalities. Both beta-arbutin and 6′-*O*-caffeoylarbutin were characterized using NMR (^1^H and ^13^C) and spectroscopic data can be found in the Appendix A. Based on these findings, we hypothesized that 6′-*O*-caffeoylarbutin shares a similar mechanism of melanin inhibition with beta-arbutin, which inhibits melanin formation by suppressing the activity of tyrosinase. The enhanced anti-melanin activity of 6′-*O*-caffeoylarbutin can be attributed to its diphenol structure. Furthermore, our previous experiments confirmed that 6′-*O*-caffeoylarbutin effectively inhibits both the monophenolase and diphenolase activities of mTyr, with IC_50_ values of 1.114 ± 0.035 μM and 95.198 ± 1.117 μM, respectively. In comparison, beta-arbutin selectively inhibits the monophenolase activity of mTyr with an IC_50_ of 8.4 mM, while having no inhibitory effect on the diphenolase activity. Notably, 6′-*O*-caffeoylarbutin exhibits stronger inhibition of the diphenolase activity of mTyr compared to the positive control kojic acid (Appendix A) [15,16].

To further validate our hypothesis, the inhibitory effect of 6′-*O*-caffeoylarbutin on the diphenolase activity of mTyr was measured by analyzing the oxidation of L-DOPA to DOPA quinones using HPLC. As shown in Figure 2, the oxidation products of L-DOPA significantly decrease with the addition of 6′-*O*-caffeoylarbutin, indicating the suppression of enzyme activity. However, the products do not disappear completely, suggesting that the enzyme activity is not completely inhibited. Therefore, 6′-*O*-caffeoylarbutin is a better inhibitor of mTyr compared to beta-arbutin and kojic acid, and its stronger inhibitory effect on the diphenolase activity of the enzyme may be the main reason for its approximately two-fold higher anti-melanin activity compared to beta-arbutin.

#### 2.1.2. mTyr Inhibition Kinetics of 6′-*O*-Caffeoylarbutin

Enzyme inhibition can be categorized into two main types, namely reversible and irreversible. To determine the nature of inhibition, the relative reaction rates of enzyme-catalyzed reactions at different inhibitor concentrations can be plotted. Figure 3A illustrates a set of straight lines intersecting at the origin when plotting the relative reaction rate against enzyme concentration under various concentrations of 6′-*O*-caffeoylarbutin. The figure clearly shows that, as the concentration of 6′-*O*-caffeoylarbutin increases, the slope of the lines gradually decreases and eventually intersects at the origin. This indicates that the inhibitory effect of 6′-*O*-caffeoylarbutin on mTyr is a reversible process, and it does not decrease the total amount of mTyr but rather inhibits its catalytic activity.

Continuing from the Lineweaver_Burk analysis, the inhibition type of 6′-*O*-caffeoylarbutin on mTyr was determined. The inhibition types include competitive inhibition, non-competitive inhibition, uncompetitive inhibition, and mixed-type inhibition, which can be determined by plotting the reciprocal of enzyme reaction rate [1/V] against the reciprocal of substrate concentration [1/S] at different inhibitor concentrations. As shown in Figure 3B, with increasing concentrations of 6′-*O*-caffeoylarbutin, the slope of the straight lines gradually increased, and, ultimately, the double reciprocal curves of enzyme reaction rate and substrate concentration intersected on the y-axis. This indicates that 6′-*O*-caffeoylarbutin inhibits mTyr through competitive inhibition. In other words, 6′-*O*-caffeoylarbutin can bind to the free tyrosinase, competing with the substrate L-DOPA for the binding site of enzyme, thereby inhibiting melanin production. The inhibition constant (Ki) for the binding of the inhibitor to the free enzyme can be calculated from the slope of the double reciprocal curves of different inhibitor concentrations against concentration. The Ki of 6′-*O*-caffeoylarbutin on mTyr was determined to be 1.34 ± 0.92 μM indicating that 6′-*O*-caffeoylarbutin affects the activity of mTyr by binding to the free enzyme, and it likely interacts with only one or a specific class of binding sites [17].

#### 2.1.3. The Chelation of 6′-*O*-Caffeoylarbutin with Copper (II) Ion in mTyr

mTyr is a binuclear copper-containing glycoenzyme that exhibits two distinct catalytic activities. The active center of mTyr is occupied by copper (II) ions, which play a crucial role in the catalytic cycles by interacting with substrates (Figure 4A). Investigating the interaction between small molecular inhibitors and copper (II) ions can be achieved through significant blue or red shifts in the characteristic absorption bands, making UV_vis spectrometry a valuable tool for studying conformational changes in proteins within enzyme_inhibitor complexes.

Figure 4A presents the UV_vis absorption spectra of a solution containing 6′-*O*-caffeoylarbutin in the presence of mTyr and CuSO_4_ separately. 6′-*O*-caffeoylarbutin exhibits two characteristic absorption bands: band I at 323 nm, corresponding to n→π* transitions within the acyl group, and band II at 291 nm, corresponding to π→π* transitions within the aromatic group. Upon the addition of copper (II) ion to the 6′-*O*-caffeoylarbutin solution, band I undergoes a large red shift to 370 nm, while band II experiences a blue shift to 270 nm. These shifts suggest an interaction between the compound and copper (II) ion, as neither the ligand nor the metal absorbs at these specific wavelengths. 

Interestingly, the addition of mTyr to the solution does not result in a change in the maximum absorption at 328 nm, corresponding to the acyl group. However, the characteristic band for the aromatic group at 290 nm exhibits a slight blue shift to 286 nm, accompanied by an increase in intensity after incubation with mTyr. This phenomenon indicates a twist in the planar structure of 6′-*O*-caffeoylarbutin following complex formation, rather than direct chelation with the copper (II) ion located in the catalytic active center of the enzyme. The interaction of 6′-*O*-caffeoylarbutin with the aromatic group suggests its involvement in the nucleophilic attack at the active site of mTyr, resulting in the observed decrease in catalytic activity.

In summary, the UV_vis absorption spectra provide valuable insights into the interaction between 6′-*O*-caffeoylarbutin, mTyr, and copper (II) ions. The observed shifts in absorption bands and changes in intensity indicate the formation of complexes and conformational changes in 6′-*O*-caffeoylarbutin upon binding to mTyr. These findings contribute to our understanding of the inhibitory effects of 6′-*O*-caffeoylarbutin on the catalytic activity of mTyr.

#### 2.1.4. The Fluorescence Quenching Effect of 6′-*O*-Caffeoylarbutin on mTyr

Fluorescence quenching refers to the phenomenon in which the intensity of fluorescence decreases as a result of interactions between molecules and fluorescent substances. These interactions can involve excited-state reactions, molecular rearrangements, energy transfer, formation of ground-state complexes, and collisional quenching [18,19].

In Figure 4B, curve 1 represents the fluorescence emission spectrum of mTyr. The intrinsic fluorescence of mTyr primarily originates from tyrosine residues at 320 nm and tryptophan residues at 335 nm. Binding of small molecules to mTyr induces changes in the microenvironment of the amino acid residues, resulting in alterations in the fluorescence intensity [20]. Curve 10 corresponds to the fluorescence emission spectrum of 6′-*O*-caffeoylarbutin, while curves 2–9 represent the fluorescence emission spectra of 6′-*O*-caffeoylarbutin at concentrations of 5, 10, 15, 25, 50, 75, 90, and 100 μM, respectively. 

The results demonstrate a significant decrease in the fluorescence intensity of mTyr in the presence of 6′-*O*-caffeoylarbutin, indicating the quenching of the intrinsic fluorescence of mTyr. Moreover, the fluorescence intensity shows a concentration-dependent reduction. As observed in Figure 4B, with increasing concentrations of 6′-*O*-caffeoylarbutin, the fluorescence intensity of mTyr gradually decreases. At a concentration of 25 μM 6′-*O*-caffeoylarbutin, the peak of the fluorescence emission spectrum of mTyr experiences a reduction of 37.18% (Appendix A). Further increasing the concentration to 100 μM results in a decrease of 12.57% in the fluorescence emission spectrum of mTyr. These findings suggest that 6′-*O*-caffeoylarbutin induces quenching of the intrinsic fluorescence of mTyr, indicating a potential interaction between 6′-*O*-caffeoylarbutin and mTyr.

According to Kim et al. [21], hydroxyl groups in inhibitors primarily contribute to quenching the intrinsic fluorescence of enzymes. Therefore, it can be inferred that 6′-*O*-caffeoylarbutin predominantly quenches the fluorescence of mTyr by interacting with the hydroxyl group on its phenyl ring.

Additionally, as the concentration of 6′-*O*-caffeoylarbutin increases, the emission peak of tryptophan residues in tyrosinase shifts from 335 nm to 355 nm, indicating a significant redshift. Similarly, the emission peak of tyrosine residues at 320 nm shifts to 310 nm, demonstrating a noticeable blueshift. This phenomenon may be attributed to the relatively hydrophobic nature of the tryptophan and tyrosine amino acid residues in mTyr. The fluorescence spectra of these residues are sensitive to changes in the surrounding microenvironment. Studies have shown that a blue shift in the emission spectrum suggests alterations in the microenvironment near tryptophan residues, while a red shift corresponds to changes in the microenvironment near tyrosine residues [22]. Therefore, it can be inferred that an interaction occurs between mTyr and 6′-*O*-caffeoylarbutin, resulting in a decrease in polarity around the tryptophan residues in mTyr, leading to a more hydrophobic environment [23]. These observations indicate that the interaction between 6′-*O*-caffeoylarbutin and mTyr induces conformational changes in the enzyme.

Fluorescence quenching can be classified into static quenching and dynamic quenching. Static quenching occurs when quenching agents bind to fluorescent substances, forming complexes that result in quenching. Dynamic quenching, on the other hand, primarily arises from thermal motion and molecular collisions. In this study, the quenching results were analyzed using the Stern_Volmer equation, as shown in Appendix A. The plot of F0/F against the concentration of 6′-*O*-caffeoylarbutin yielded a straight line with a good linear relationship (R^2^ = 0.9802). The slope of this line can be used to calculate the quenching rate constant Kq, which was determined to be 4.11 × 10^12^ L·(mol·s)^−1^. Typically, the threshold value for the maximum collisional quenching rate constant between small molecular quenchers and large biomolecules is 2.0 × 10^10^ L·(mol·s)^−1^. If the calculated quenching rate constant is below this threshold, it indicates a dynamic quenching mechanism. Conversely, if the quenching rate constant exceeds this threshold, the quenching process may involve static quenching [23]. In the case of 6′-*O*-caffeoylarbutin and mTyr, the Kq significantly exceeds the threshold for the maximum collisional quenching rate constant, suggesting that the fluorescence quenching process of mTyr induced by 6′-*O*-caffeoylarbutin is primarily attributed to the formation of a complex between 6′-*O*-caffeoylarbutin and mTyr, indicating that static quenching is the predominant mechanism involved in the fluorescence quenching process.

In the static quenching process, the relationship between fluorescence intensity and quencher concentration can be described by Equation (1).
lg[(F_0_ − F)/F] = lgKa + nlg[Q](1)
where F_0_ and F represent relative fluorescence intensities of the solution without and with the quencher, respectively. Ka represents the apparent binding constant between the enzyme and the quencher, n represents the number of binding sites on each biomacromolecule for the quencher, and [Q] denotes the concentration of the quencher. In this study, a double-logarithmic linear regression analysis of lg[(F_0_ − F)/F] against lg[Q] was conducted to evaluate the interaction between 6′-*O*-caffeoylarbutin and mTyr. Appendix A illustrates the resulting regression line, which provided a slope and intercept for the calculation of Ka and n. The obtained values were determined as (10.79 ± 0.82) × 10^4^ L∙mol^−1^ and 0.79 ± 0.06, respectively. These findings indicate the formation of a stable complex between 6′-*O*-caffeoylarbutin and mTyr, involving a single binding site. 

### 2.2. Computational Simulation of Interaction between 6′-O-Caffeoylarbutin and mTyr

#### 2.2.1. Molecular Docking

Based on the molecular docking results, it was found that 6′-*O*-caffeoylarbutin exhibits a favorable binding affinity with mTyr, with a docking score of -8.0 kcal∙mol^−1^, whereas the docking score of kojic acid was −5.7 kcal∙mol^−1^ (Appendix A). This indicates a strong interaction between 6′-*O*-caffeoylarbutin and the receptor. Figure 5A demonstrates the interaction mode between 6′-*O*-caffeoylarbutin and the target protein mTyr, showing that 6′-*O*-caffeoylarbutin effectively enters the active pocket of mTyr protein to exert its activity.

Further analysis focuses on the interaction between 6′-*O*-caffeoylarbutin and key amino acid residues in the active pocket of mTyr. Figure 5B illustrates the strong interaction of 6′-*O*-caffeoylarbutin with the active site residues, which are not directly involved in the interaction with the Cu(II) ion of mTyr, consistent with the UV_vis spectra results [24]. Specifically, the 4-hydroxyphenoxy group of 6′-*O*-caffeoylarbutin forms hydrogen bonds with Glu322 (3.23 Å) and His85 (5.33 Å), while the hydroxyl groups in the glycosidic bond form two hydrogen bonds with Val 283 (4.48 Å) and Gly 281 (3.91 Å). An additional hydrogen bond is formed between the oxygen atom of the acyl group and His 85 (5.33 Å). These five hydrogen bonds contribute to the conformational stability of the receptor protein and result in the inhibition of mTyr catalytic activity.

Furthermore, the complex formation between 6′-*O*-caffeoylarbutin and mTyr involves three stable hydrophobic interactions (Appendix A). One is the pi-alkyl interaction between 4-hydroxyphenoxy and Ala286 (6.75 Å), as well as between 3,4-hydroxyphenoxy and Val283 (5.51 Å). Another interaction is the pi_pi stacked interaction formed by the aromatic ring of 4-hydroxyphenoxy with His263 (4.78 Å). Additionally, a pi_sigma interaction is formed between 4-hydroxyphenoxy and Val283 (4.29 Å). Notably, Val283 plays a crucial role in the interaction between mTyr protein and 6′-*O*-caffeoylarbutin, as it is involved in both hydrogen bonding and hydrophobic interactions, including pi_sigma and pi_alkyl interactions.

Moreover, 6′-*O*-caffeoylarbutin predominantly interacts with 15 amino acid residues through van der Waals forces. Among them, four hydrophobic residues, namely Val248, Phe90, Phe264, and Phe292, located at the periphery of the binding cavity, contribute to its hydrophobic nature. The incorporation of the caffeoyl group into beta-arbutin significantly alters the binding interactions between the protein and ligand, thereby enhancing the targeting ability of 6′-*O*-caffeoylarbutin towards mTyr. The docking results demonstrate that 6′-*O*-caffeoylarbutin induces conformational changes in the active site, impairing the functionality of the enzyme. The dominant conformation of 6′-*O*-caffeoylarbutin can be regarded as an inhibitor, as the formation of the enzyme_inhibitor complex impedes the release of oxidation products and the entry of L-DOPA. This finding provides a plausible explanation for the diphenolase inhibitory activity of 6′-*O*-caffeoylarbutin, which is consistent with the results obtained from HPLC analysis.

#### 2.2.2. MD Simulation

The root mean square deviation (RMSD) of the MD simulation trajectory is an essential criterion for evaluating the stability of protein_ligand complexes and the accuracy of docking results [25]. As depicted in Figure 6A, the RMSD analysis of the 6′-*O*-caffeoylarbutin and mTyr protein complex reveals an initial increasing trend in the RMSD of mTyr protein during the early stages of the simulation. After approximately 10 ns, the trajectory stabilizes, with two significant fluctuations occurring around 20 ns and 30 ns, resulting in a maximum RMSD fluctuation of 0.22 nm for mTyr protein. Throughout the rest of the simulation, the RMSD values of mTyr protein fluctuate narrowly around 0.18 nm. In comparison, the complex reaches a relatively stable state after 22 ns, with an RMSD value fluctuating around 0.20 nm.

The RMSD values of the complex are close to those of mTyr protein but with overall smaller fluctuations, providing compelling evidence for the stable existence of the 6′-*O*-caffeoylarbutin-mTyr complex.

Root mean square fluctuation (RMSF) measures the magnitude of atomic deviations relative to their average positions, indicating the flexibility and local motion characteristics of the system [26]. From Figure 6B, it can be observed that, among the 392 amino acid residues, the mTyr protein exhibits higher RMSF values mainly in the peptide chain region of 240–300, indicating higher flexibility in that area. However, upon binding of the ligand 6′-*O*-caffeoylarbutin to the mTyr protein, the RMSF values of 227 amino acid residues increase, while the RMSF values of 165 amino acid residues decrease. Particularly within the range of 240–300, the RMSF values of amino acid residues in the complex are significantly lower than those of mTyr protein. This suggests a strong binding capacity between the amino acid residues in that region and the ligand compound, forming stable hydrogen bonds or hydrophobic interactions. This interconnection between the ligand and the surrounding amino acids in the binding pocket contributes to the formation of a stable complex, resulting in a substantial reduction in RMSF values compared to mTyr protein [27,28].

Further analysis was conducted to investigate the reasons for the conformational flexibility changes in mTyr protein, and various dynamic geometric properties of both the protein and the complex were computed using the equilibrium trajectories of the two simulation systems. These properties include the number of hydrogen bonds (NHB), radius of gyration (Rg), and solvent accessible surface area (SASA). As shown in Appendix A, 6′-*O*-caffeoylarbutin initially forms four hydrogen bonds with mTyr protein, and the number of hydrogen bonds fluctuates between one and six during the simulation. Stable hydrogen bonds were formed between 6′-*O*-caffeoylarbutin and mTyr protein, particularly with Gly281, His85, His244, and Val283, with occupancy rates of 19.3%, 8.6%, 6.5%, and 6.2%, respectively. These results indicate the crucial role played by these hydrogen bonds in the interaction between 6′-*O*-caffeoylarbutin and mTyr protein, suggesting that they are key factors contributing to the inhibition of mTyr protein activity by 6′-*O*-caffeoylarbutin.

Rg is a measure of protein structure compactness and conformational flexibility, where a higher Rg value indicates a looser protein structure [29]. From Appendix A, the Rg value of the complex is higher than that of mTyr protein, indicating that the inclusion of the 6′-*O*-caffeoylarbutin ligand results in a more relaxed protein structure. This may be attributed to the conformational changes induced by the hydrophobic interactions between the ligand and amino acid residues, leading to a more relaxed protein structure. SASA analysis is used to determine the protein surface area in contact with solvent molecules. Appendix A shows that the SASA value of the complex is higher than that of mTyr protein, indicating an increased external surface area of the complex, which may be related to the interactions between the ligand and the protein. The ligand entering the active pocket of mTyr protein and binding to the surrounding amino acid residues lead to an increased exposed surface area of the protein, thereby increasing the SASA value of the complex.

In summary, the hydrogen bonds and hydrophobic interactions between the 6′-*O*-caffeoylarbutin ligand and mTyr protein contribute to a more relaxed protein structure (higher Rg) and an increased surface area (higher SASA). These structural changes are likely significant factors contributing to the inhibition of mTyr protein activity by 6′-*O*-caffeoylarbutin. The aforementioned results provide substantial evidence supporting the accuracy of molecular docking and MD simulations.

### 2.3. Safety Evaluation

#### 2.3.1. Cellular Assays

In this study, the CCK-8 assay was used to investigate the effects of 6′-*O*-caffeoylarbutin on cell proliferation and cytotoxicity in A375 and HaCaT cells. The results revealed IC_50_ values of 313.001 μM and >400 μM for A375 and HaCaT cells, respectively, indicating low toxicity of 6′-*O*-caffeoylarbutin towards both cell types. As shown in Figure 7A,B, minimal cytotoxicity of 6′-*O*-caffeoylarbutin was observed within the concentration range of 3.125–100 μg∙mL^−1^, with inhibition rates below 5% for A375 cells and below 3% for HaCaT cells, and no significant morphological changes were observed. At a concentration of 200 μg∙mL^−1^, 6′-*O*-caffeoylarbutin exerted a stronger inhibitory effect on A375 cells, resulting in an inhibition rate of 20% and distinct morphological alterations characterized by reduced cell_cell contacts, chromatin aggregation, and crescent-shaped structures along the nuclear membrane. In contrast, at the same concentration, 6′-*O*-caffeoylarbutin exhibited a modest inhibitory effect of only 3.6% on HaCaT cells, without significant morphological changes. Furthermore, when the concentration of 6′-*O*-caffeoylarbutin was increased to 400 μg∙mL^−1^, it demonstrated a more pronounced inhibitory effect on A375 cells, exhibiting an inhibition rate of 67.8% and significant morphological changes, including loss of cell_cell contacts, formation of membrane blebs, and the presence of individual apoptotic bodies. Conversely, its inhibitory effect on HaCaT cells was weaker, with an inhibition rate of only 14.5%, and only slight changes in cell morphology were observed. These findings suggest that 6′-*O*-caffeoylarbutin exhibits favorable cellular safety at lower concentrations and limited toxicity towards HaCaT cells even at higher concentrations, while selectively inhibiting the proliferation of A375 cells.

#### 2.3.2. Acute Oral Toxicity Assays 

Further validation of the safety of 6′-*O*-caffeoylarbutin was achieved through animal experiments. During the 14-day acute toxicity experiment, the toxic effects of 6′-*O*-caffeoylarbutin on the general symptoms of different groups were assessed (Table 1). No significant adverse effects were observed in the Vehicle or CA-3 groups. However, when the single dose was increased to 23848 mg·kg^−1^ (CA-2 group), one male mouse showed reduced activity. At an oral administration dosage of 28056 mg·kg^−1^ (CA-1 group), eight mice exhibited reduced activity, two male mice developed a prone state symptom, and four mice experienced shortness of breath. These symptoms mainly occurred on the day of and the first day after oral administration, and all animals returned to normal by the third day.

Similarly, in terms of mortality, no deaths were observed in the Vehicle or CA-3 groups during the experiment (Appendix A). All six deaths occurred in either the CA-1 or CA-2 group, with five deaths in the high-dose CA-1 group (23848 mg·kg^−1^) and one death in the medium-dose CA-2 group (20272 mg·kg^−1^). The mortality rate and toxic symptoms were positively correlated with dosage, with males showing slightly more pronounced symptoms than females.

Furthermore, histological analysis was performed on the six mice that died during the experiment. Dark red discoloration of the liver or lower margin of the liver was observed in a few mice from the CA-1 (28,056 mg·kg^−1^) group, while no structural damage was observed in the others. After the 14-day acute toxicity test, all animals were sacrificed for histological analysis, and no abnormalities were observed in the internal organs.

Additionally, on the third day after administration, the mean body weight of male mice in the CA-1 and CA-3 groups was slightly higher than that of the Vehicle group. However, at other time points, the body weight of all dose groups was slightly lower than that of the Vehicle group, with statistically significant differences between groups (*p* < 0.05/0.01). On the first day after oral administration, the average weight of mice treated with 6′-*O*-caffeoylarbutin showed a slight increase compared to the Vehicle group, with statistically significant differences (*p* < 0.05/0.01). As the observation period progressed, the average weight of the CA-1/CA-2/CA-3 groups approached or slightly exceeded that of the Vehicle group (Appendix A). By the 14th day, the average body weight of mice treated with 6′-*O*-caffeoylarbutin was comparable to that of the Vehicle group, while male mice had slightly higher weights than the Vehicle group, indicating that female weight gain was slightly slower than that of males in the later stage. 

Therefore, these comprehensive findings underscore the potential of 6′-*O*-caffeoylarbutin as a selective tyrosinase inhibitor with low toxicity and encourage its further exploration in skincare and food preservation applications.

### 2.4. A Preliminary Investigation of the Browning Inhibitory Effect of 6′-O-Caffeoylarbutin on Apple Juice

The preservation effect of different concentrations of 6′-*O*-Caffeoylarbutin on apple juice is illustrated. As shown in Appendix A, 6′-*O*-Caffeoylarbutin demonstrated a significant reduction in the formation of browning compounds in a dose-dependent manner compared to the apple juice blank control, which lacked a browning inhibitor. Browning in fruits and vegetables is typically caused by the catalytic conversion of tyrosine to dopa quinone by the enzyme tyrosinase. Considering the inhibitory effect of 6′-*O*-Caffeoylarbutin on mTyr, it can be inferred that this compound effectively suppresses the activity of polyphenol oxidase in freshly squeezed apple juice. As a result, it slows down the oxidation process of phenolic compounds, thereby inhibiting the occurrence of browning in apple juice. This indicates that 6′-*O*-Caffeoylarbutin has a positive impact on preserving the quality and appearance of apple juice by minimizing browning reactions.

## 3. Materials and Methods

### 3.1. Reagents

The 6′-*O*-caffeoylarbutin standard (purity > 90%) was isolated and purified from Quezui tea by our laboratory [19]. Mushroom tyrosinase (mTyr, EC 1.14.18.1) was obtained from Sigma-Aldrich (St. Louis, MO, USA). L-tyrosine and 3,4-dihydroxy L-phenylalanine (L-DOPA) were procured from Adamas-Beta Ltd. (Shanghai, China). Ultrapure water was used throughout all the experiments. All other chemicals and solvents were sourced from Shanghai Titan Scientific Co., Ltd. (Shanghai, China), unless otherwise specified. All chemicals used were of analytical grade or better, and were used directly in the experiments without further purification.

### 3.2. mTyr Inhibition Mechanism of 6′-O-Caffeoylarbutin

#### 3.2.1. Determination of Inhibitory Effect of 6′-*O*-Caffeoylarbutin on Oxidation of L-DOPA Catalyzed by mTyr

The inhibitory effect of 6′-*O*-caffeoylarbutin on the oxidation of L-DOPA catalyzed by mTyr was studied using Agilent 1200 series HPLC analysis (Palo Alto, CA, USA), with minor modifications to the previous procedure [21]. The experiment was conducted with 0.5 mM L-DOPA as a diphenolase substrate for 31.25 U·mL^−1^ mTyr in the absence of an inhibitor. Subsequently, 2 μg·mL^−1^ of 6′-*O*-caffeoylarbutin was added, and the reaction mixture was incubated for 0, 15, and 30 min at 30 °C. Samples (10 μL) were injected after filtration onto the HPLC system equipped with a Capcell pak C18 column (250 × 4.6 mm, 5 μm, Shiseido, Tokyo, Japan) operating at a flow rate of 1.0 mL·min^−1^. The column temperature maintained at 30 °C and the detection wavelength was set at 280 nm. The mobile phase was consisted of water (A) and methanol (B), and a gradient elution employed with the following linear gradient for solvent B: 5% (0 min), 20% (10 min), 50% (20 min), 80% (25 min), and 100% (35 min).

#### 3.2.2. Determination of mTyr Inhibition Kinetics of 6′-*O*-Caffeoylarbutin

The mTyr inhibition kinetics of 6′-*O*-caffeoylarbutin were studied using a method similar to the previously reported procedure for determining the inhibitory activity of 6′-*O*-caffeoylarbutin on mTyr [30]. 

I. Enzyme inhibition mechanism study. For the investigation of the enzyme inhibition mechanism, L-DOPA was used at a final concentration of 0.5 mmol·L^−1^, and the mTyr was set at final concentrations of 8.1, 32.5, and 65.0 U·mL^−1^. The rate of change of optical density (OD) at 490 nm was measured for various concentrations of 6′-*O*-caffeoylarbutin (ranging from 10 to 100 μM). A graph was plotted to depict the relationship between the rate of change of OD values and the mTyr concentration under the influence of different concentrations of 6′-*O*-caffeoylarbutin.

II. Enzyme inhibition type study. To determine the enzyme inhibition type, L-DOPA was used with final concentrations of 0.2, 0.5, 0.6, and 0.8 mmol·L^−1^, while mTyr was used at a final concentration of 31.25 U·mL^−1^. The rate of change of OD at 490 nm was measured for various concentrations of 6′-*O*-caffeoylarbutin (ranging from 0 to 100 μM). A reciprocal plot was generated using the rate of change of OD values and L-DOPA concentration. The inhibition constants were obtained based on the inhibition type of 6′-*O*-caffeoylarbutin on mTyr using quadratic plotting of the 6′-*O*-caffeoylarbutin concentration and the slope and vertical intercept of the line in the Lineweaver_Burk double reciprocal plot and Dixon plot [31,32].

#### 3.2.3. Determination of the Chelation of 6′-*O*-Caffeoylarbutin with Copper (II) Ion in mTyr

The chelation of 6′-*O*-caffeoylarbutin with the copper (II) ion active center of mTyr was determined using a UV-2600 spectrophotometer (Shimadzu, Kyoto, Japan). The absorption spectra were recorded in the wavelength range of 240 nm to 400 nm at 30 °C, following the literature with minor modifications [33]. To perform the assay, a 10 μM solution of 6′-*O*-caffeoylarbutin in phosphate-buffered saline (PBS) was mixed separately with a 31.25 U·mL^−1^ mTyr solution in PBS and a 125 μM copper sulfate (CuSO_4_) aqueous solution. The final volume of all working solutions was adjusted to 3 mL using PBS, and this volume was kept constant for all samples. After gently mixing the solutions using a vortex shaker, the samples were incubated at 30 ± 2 °C for 10 min with occasional shaking. Following the incubation period, the absorption spectra were recorded using UV-vis spectrophotometer. This experimental procedure allowed for the determination of the chelation of 6′-*O*-caffeoylarbutin with the copper (II) ion active center of mTyr through the analysis of the absorption spectra.

#### 3.2.4. Determination of the Fluorescence Quenching Effect of 6′-*O*-Caffeoylarbutin on mTyr

The fluorescence quenching role of 6′-*O*-caffeoylarbutin on mTyr was investigated using a RF-6000 fluorescence spectrophotometer (Shimadzu, Kyoto, Japan) with a xenon flash lamp as a light source, according to the literature with minor alterations [34]. The excitation wavelength was 280 nm, with a scanning wavelength ranging from 290 to 500 nm at 37 °C. The excitation and emission bandwidths were 2.5 nm and 5.0 nm, respectively. A 3 mL mixture containing 2.0 mL of 31.25 U·mL^−1^ mTyr solution, 0.5 mL of 6′-*O*-caffeoylarbutin solution (with 3% dimethyl sulfoxide) at concentrations from 0 to 100 μM, and 0.5 mL of PBS was used to collect the fluorescence intensity curves. The PBS served as the corresponding blank to correct the fluorescence intensities. The fluorescence quenching mechanism between mTyr and 6′-*O*-caffeoylarbutin was described using the Stern-Volmer Equation (2) [21,32,35].
(2)F0F=1+KSVQ=1+Kqτ0[Q]
where *K_SV_* is the Stern-Volmer quenching constant, [*Q*] represents the concentration of 6′-*O*-caffeoylarbutin, *K_q_* is bimolecular quenching rate constant, and $\tau_{0}$ denotes the average lifetime of the fluorescence molecules in the absence of 6′-*O*-caffeoylarbutin (value is 10^−8^ s).

### 3.3. Computational Simulation of Interaction between 6′-O-Caffeoylarbutin and mTyr

#### 3.3.1. Molecular Docking 

The crystal structure of mTyr (PDB ID: 2Y9X) was obtained from the Protein Data Bank (PDB), and water molecules, original ligands, and redundant amino acid chains were manually removed. The crystal structure of the ligand molecule, 6′-*O*-caffeoylarbutin, was sourced from the PubChem Project and saved in PDBQT format using PyMOL, serving as the initial conformation for molecular docking. For docking experiments, the AutoDock Vina 1.1.2 software package was utilized to prepare the 6′-*O*-caffeoylarbutin and mTyr. The MM2 small molecule force field was applied to perform geometric optimization on 6′-*O*-caffeoylarbutin, while the Kollman charge and polar hydrogen atoms were added to mTyr. During the docking process, the receptor grid box size was set to 25 × 25 × 25, the docking model output was set to 20, and other parameters were kept at their default settings [33,36]. A docking experiment was performed between 6′-*O*-caffeoylarbutin and mTyr using the aforementioned parameters. Subsequently, the obtained ligand conformations were analyzed for their interactions and binding energies. Based on the bonding interactions and the lowest affinity values, reasonable docking results were selected as the initial conformations for molecular dynamics (MD) simulations.

#### 3.3.2. MD Simulation

MD simulations of the 6′-*O*-caffeoylarbutin and mTyr complex system were conducted for 50 ns using GROMACS 2020.4 software, employing the amber14sb force field. The topologies of the small molecules were generated using sobtop_1.0 (Tian Lu, Sobtop, Version 1.0, http://sobereva.com/soft/Sobtop, accessed on 4 May 2023). The complex system was solvated using the SPC water model with periodic boundary conditions. Sodium or chloride counter ions were added to neutralize each system. Energy minimization was carried out using the steepest descent algorithm until a tolerance value of 5 kJ·mol^−1^·nm^−1^ was achieved. Following energy minimization, the protein molecules underwent positional constraint equilibration for 100 ps, utilizing a combination of NVT (constant number of particles, volume, and temperature) and NPT (constant number of particles, pressure, and temperature) ensembles with standard coupling methods. Subsequently, trajectories were generated with a time step of 2 fs and simulated for 50 ns in production MD, with frames saved every 10 ps. The MM/PB (GB) SA method was employed to calculate the combined free energy using the equilibrium trajectory.

### 3.4. Safety Evaluation

#### 3.4.1. Cellular Assays

I. Cell lines and cell culture. Human melanoma A375 tumor cells and human immortalized keratinocyte (HaCaT) non-tumor cells were obtained from Nanjing Pusheng Biomedical Technology Co., Ltd. (Nanjing, China). The cells (A375/HaCaT) were cultured in Dulbecco’s modified Eagle’s medium (DMEM, Gibco, New York, NY, USA) supplemented with 10% (*v*/*v*) fetal bovine serum (FBS, Gibco, New York, NY, USA), and incubated at 37 °C with 5% CO_2_ in a humidified incubator.

II. Cell viability assay. Cell proliferation and cytotoxicity assays were conducted using the Cell counting kit-8 (CCK-8) on A375 and HaCaT cells. Briefly, the cells were initially seeded in 96-well plates at a density of 4 × 10^4^ cells per well and incubated for 24 h at 37 °C with 5% CO_2_ in a humidified incubator. Each treatment was performed in six duplicate wells. The tested compounds were added at various concentrations, and the cells were then incubated for 72 h. Following the incubation period, CCK-8 solution (Dojindo Laboratories, Kumamoto, Japan) was added to each well, and the plates were further incubated for 2 h. The absorbance of each well at 450 nm was measured using an absorbance microplate reader (BioTek ELx800, Winooski, VT, USA). The assay included negative controls in which cells were treated with ultrapure water and positive controls in which cells were treated with 10 µg·mL^−1^ taxol. The percentage of cell growth inhibition was calculated using the following Equation (3).
(3)$$\mathrm{Cell}\;\mathrm{growth}\;\mathrm{inhibition}\;\mathrm{rate}\;[\%]=\frac{B_{1}-B_{2}}{B_{1}-B_{3}}\times 100\;$$
where *B*_1_ represents the absorbance of a mixture containing cells, medium, and CCK-8 solution, *B*_2_ represents the absorbance of a mixture containing cells, medium, CCK-8 solution, and the test compound; and *B*_3_ represents a mixture containing medium and CCK-8 solution.

Cell growth and morphology were observed using an inverted microscope (Olympus IX51, Tokyo, Japan).

#### 3.4.2. Acute Oral Toxicity Assays

I. Animals. 80 ICR mice (56 weeks, SPF), including 40 males (28.4328 g) and 40 females (20.0228 g), were used in the acute toxicity study. All animals were purchased from Beijing Vital River Laboratory Animal Technology Co., Ltd. (certificate number: No.11400700215735, Beijing, China) and were fed with a standard animal diet (certificate number: 17033313, Beijing Keao Xieli Feed Co., Ltd., Beijing, China,) and tap water during the experimental period. Male and female mice were raised separately in sterile PC transparent cages in the room with a 12 h light-dark cycle. The animal procedures followed the guidelines of the National Institute of Health Guide for the Care and Use of Laboratory Animals and were approved by the Institutional Animal Care and Use Committee (IACUC) of Yunnan Institute of Materia Medica.

II. Acute oral_toxicity. The acute toxicity test was determined based on the Technical Guidelines for the Study of Drug Toxicity by Single Dose Administration (State Food and Drug Administration (SFDA) Circular Annex 2, 2014) and the principles of OECD Guidebook 425 (OECD, 2008). In this experiment, a 5% sodium carboxymethyl cellulose solution (5% CMC-Na) was chosen as the Vehicle and the solvent for 6′-*O*-caffeoylarbutin. For the oral acute toxicity experiment, a total of 80 mice (40 males and 40 females) were divided into groups as follows: Vehicle, CA-1 (28,056 mg·kg^−1^ 6′-*O*-caffeoylarbutin), CA-2 (23,848 mg·kg^−1^ 6′-*O*-caffeoylarbutin), and CA-3 (20,272 mg·kg^−1^ 6′-*O*-caffeoylarbutin), with each group consisting of 20 mice (10 males and 10 females). Each mouse in the Vehicle group received an administration of 40 mL∙kg-1 of 5% CMC-Na. The treatment groups were orally administrated 6′-*O*-caffeoylarbutin twice within a 24_h period, with an interval of approximately 3.5–4.0 h, by dissolving in 5% CMC-Na at three different concentrations. Following administration, all animals in each group were closely monitored for about 2–5 h to observe any physical changes, monitor body weight, and record the death rate [37]. The surviving mice were observed and monitored for an additional 14 days, with their body weight recorded on the 1st, 3rd, 7th, and 14th days. Animals that died during the experimental period or those that survived until the conclusion of the toxicity test were subjected to necropsy, and their organs were histopathologically observed and analyzed. All studies were conducted in accordance with the requirements of Good Laboratory Practice.

### 3.5. A Preliminary Investigation of the Browning Inhibitory Effect of 6′-O-Caffeoylarbutin on Apple Juice

To investigate the anti-browning effect of 6′-*O*-caffeoylarbutin, apple juice was used following a modified method based on the literature [38]. Zhaotong Red Fuji apples were carefully chosen to ensure uniform shape, color, size, and maturity, free from any physical damage. After peeling, the apples were sliced into 10 g portions. These apple pieces were then immersed in solutions containing different concentrations of 6′-*O*-caffeoylarbutin for 10 min, followed by homogenization. The homogenized samples were subsequently subjected to centrifugation at 8000 rpm for 5 min at 4 °C, and the resulting supernatant (3 mL) was collected. Simultaneously, a blank control was prepared by extracting apple juice without the addition of any anti-browning agent. The absorbance of the samples at 420 nm was measured using a UV_vis spectrophotometer.

### 3.6. Data Processing and Analysis

Each data point represents the average value for a finite set of triplicate experiments on a sample. Statistical significance was assessed using a one-way analysis of variance (ANOVA) in Microsoft Excel 2019. A P value of less than 0.05 (typically ≤ 0.05) was considered statistically significant.

## 4. Conclusions

In summary, 6′-*O*-caffeoylarbutin, a phenolic glycoside abundant in the health beverage Quezui tea, has exhibited an exceptionally potent inhibitory effect on tyrosinase activity, with IC_50_ values of (1.1 ± 0.1) μM for monophenolase and (95.2 ± 1.1) μM for diphenolase. It has been validated as a reversible and competitive type inhibitor of mTyr. The interaction mechanism involves the formation of a stable 6′-*O*-caffeoylarbutin_enzyme static complex, achieved through a specific binding site to an enzyme site. By embedding into the active pocket of mTyr and forming stable hydrogen bonds and hydrophobic interactions with surrounding amino acid residues, 6′-*O*-caffeoylarbutin achieves effective binding with the enzyme. Notably, it does not bind to the active center copper (II) ion.

Furthermore, the study unveiled that 6′-*O*-caffeoylarbutin exhibits a prominent inhibitory impact on A375 cells, while manifesting lower toxicity towards HaCaT cells. This underscores its favorable cellular safety profile and selectivity. Although it manifests a certain degree of toxicity within a specific dose range, it remains devoid of inducing severe visceral damage, thereby affirming a level of safety within defined conditions. Intriguingly, 6′-*O*-caffeoylarbutin also demonstrates its prowess in preserving apples by adeptly mitigating browning in apple juice, thus auguring well for its potential utility in food preservation applications.

Although it manifests a certain degree of toxicity within a specific dose range, it remains devoid of inducing severe visceral damage, thereby affirming a level of safety within defined conditions. Intriguingly, 6′-*O*-caffeoylarbutin also demonstrates a preserving effect on apples by effectively reducing the degree of browning in apple juice, thereby offering potential application value in the field of food preservation. 

In conclusion, the findings of this study furnish indispensable theoretical and practical underpinnings for the innovation of novel tyrosinase inhibitors, while concurrently expanding the horizons for the potential applications of 6′-*O*-caffeoylarbutin in the pharmaceutical and food sectors. This significant contribution serves to propel the advancement of this field and chart a course for future exploration in this realm.

## Figures and Tables

**Figure 1 ijms-25-00972-f001:**
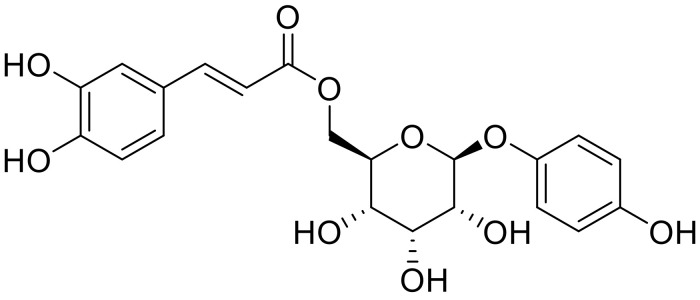
Chemical structure of 6′-*O*-caffeoylarbutin.

**Figure 2 ijms-25-00972-f002:**
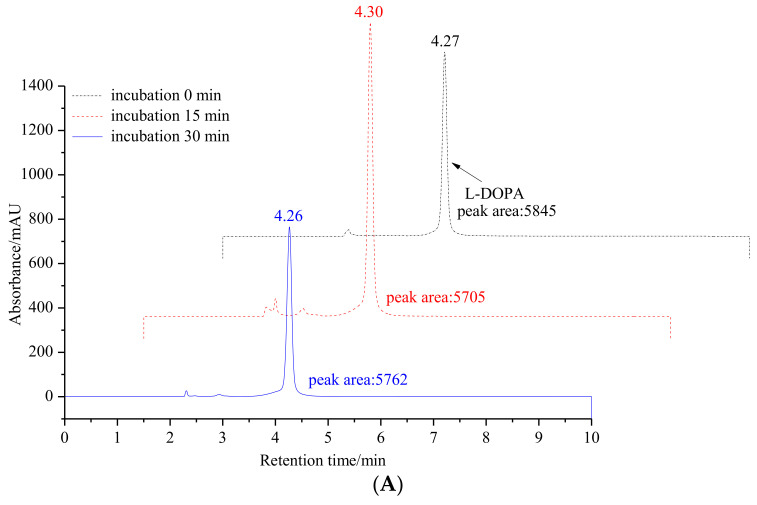
Monitoring changes in the content of L-DOPA oxidation products through HPLC analysis: (**A**). Incubation of L-DOPA as a diphenolase substrate for mTyr without 6′-*O*-caffeoylarbutin at 30 °C with incubation times of 0, 15, and 30 min. (**B**). Incubation of L-DOPA as a diphenolase substrate for mTyr in presence of 6′-*O*-caffeoylarbutin at 30 °C with incubation times of 0, 15, and 30 min. (**C**). Incubation of mTyr in the presence of 6′-*O*-caffeoylarbutin at 30 °C with incubation times of 0, 15, and 30 min at 30 °C.

**Figure 3 ijms-25-00972-f003:**
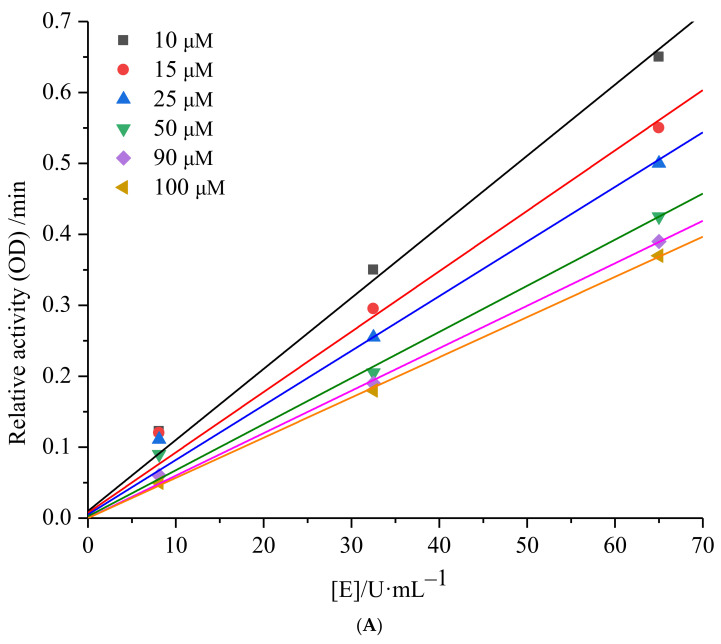
mTyr inhibition kinetics of 6′-*O*-caffeoylarbutin: (**A**). Inhibitory type of 6′-*O*-caffeoylarbutin on mTyr. (**B**). Lineweaver_Burk plots of 6′-*O*-caffeoylarbutin on mTyr.

**Figure 4 ijms-25-00972-f004:**
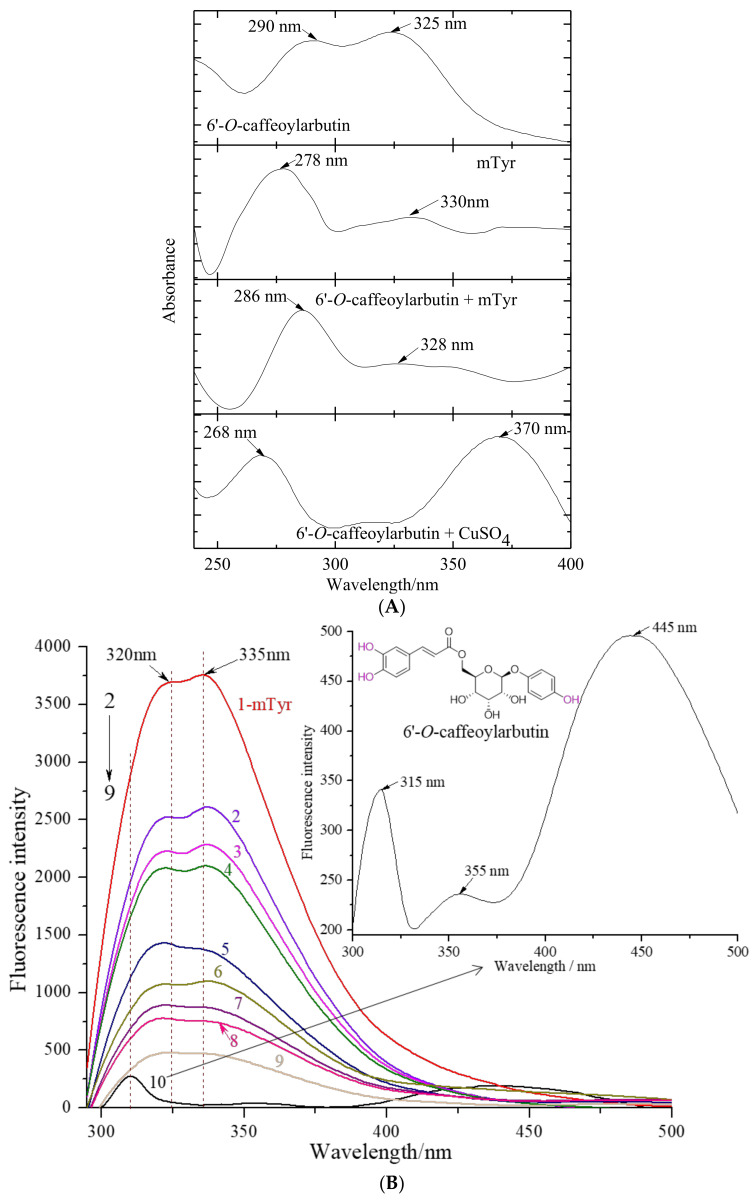
UV_vis spectra for the chelation of 6′-*O*-caffeoylarbutin with copper (II) ion in mTyr and fluorescence quenching effect of 6′-*O*-caffeoylarbutin on mTyr: (**A**). UV_vis spectra for the chelation between 6′-*O*-caffeoylarbutin and copper (II) ions. (**B**). Intrinsic fluorescence spectra of mTyr in the presence of 6′-*O*-caffeoylarbutin at different concentrations; the concentrations of 6′-*O*-caffeoylarbutin for curves 1–9 were 0, 5, 10, 15, 25, 50, 75, 90, and 100 μM, respectively; curve 10 represents the fluorescence intensity of 6′-*O*-caffeoylarbutin at the concentration of 0.02 μM.

**Figure 5 ijms-25-00972-f005:**
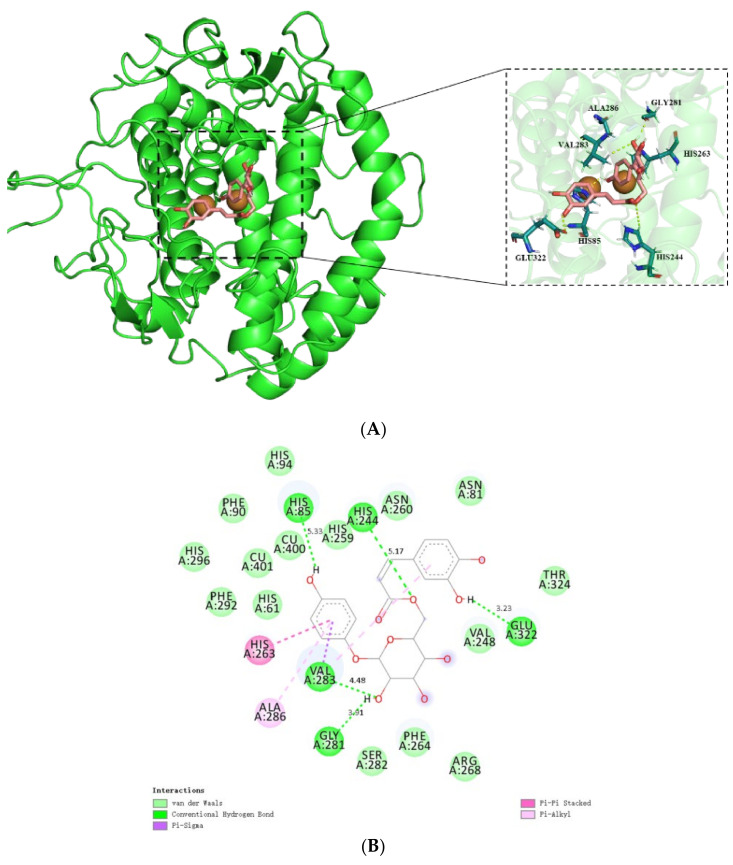
Molecular docking simulation of 6′-*O*-caffeoylarbutin with mTyr: (**A**). The docking of 6′-*O*-caffeoylarbutin with the pocket of mTyr. (**B**). The docking of 6′-*O*-caffeoylarbutin with amino acid residue of pocket of mTyr.

**Figure 6 ijms-25-00972-f006:**
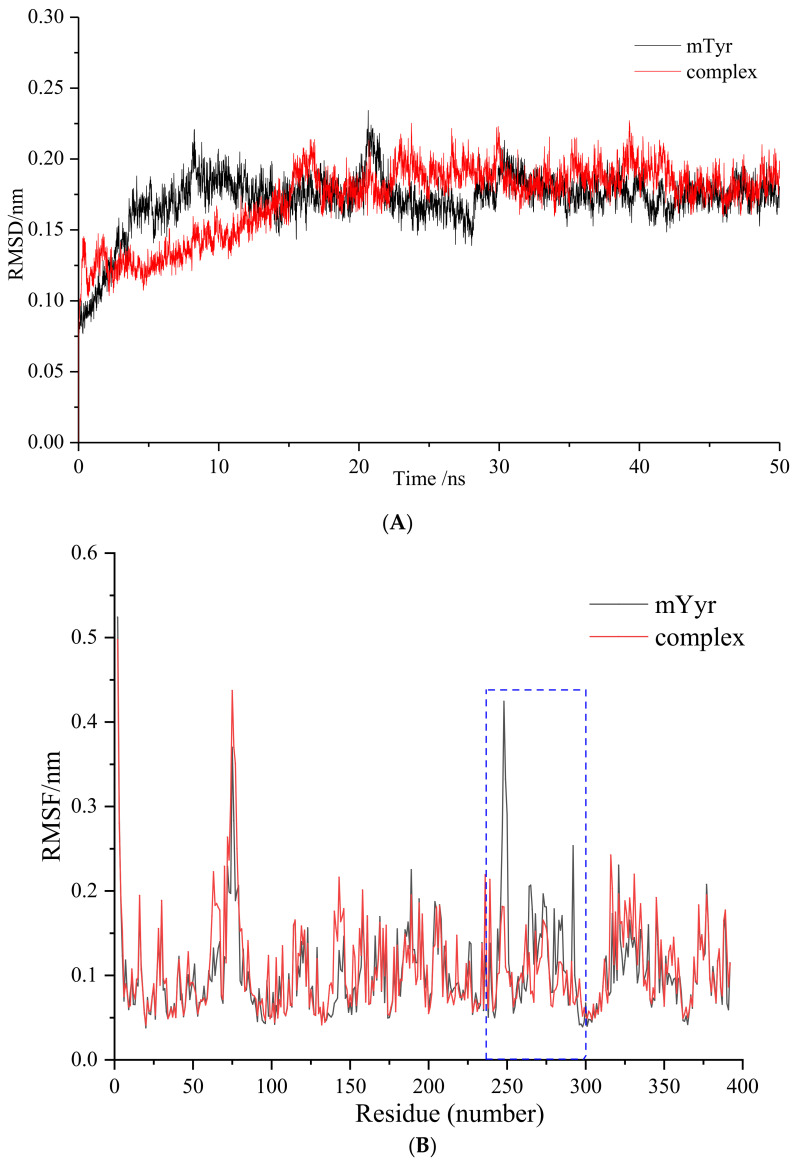
Molecular dynamics (MD) simulation of 6′-*O*-caffeoylarbutin with mTyr: (**A**). Root mean square deviation (RMSD) change curve. (**B**). Root mean square fluctuation (RMSF) change curve, blue dotted box denotes the peptide chain region of 240–300.

**Figure 7 ijms-25-00972-f007:**
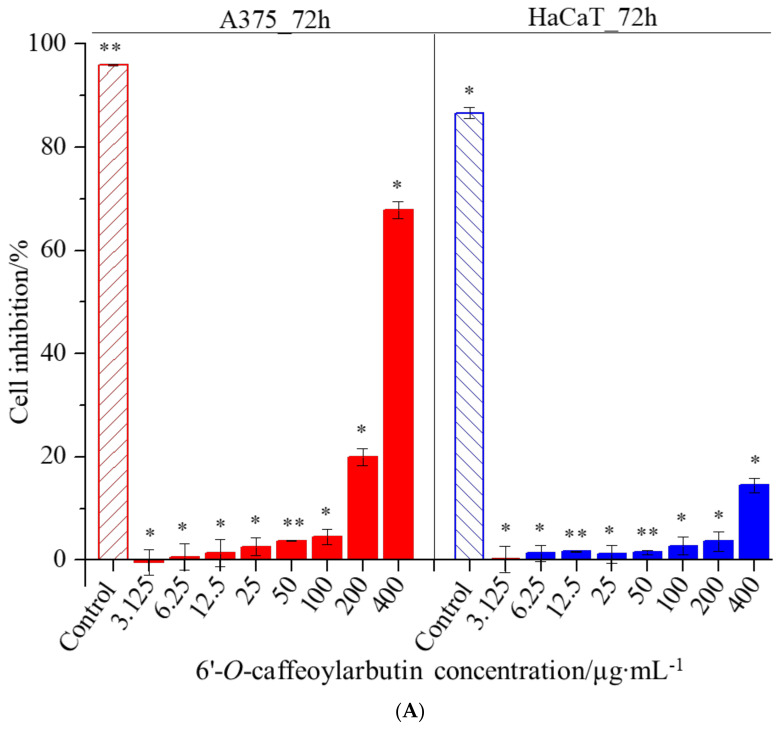
The effects of 6′-*O*-caffeoylarbutin on cell proliferation and cytotoxicity in A375 and HaCaT cells: (**A**). Dose-dependent response of 6′-*O*-caffeoylarbutin on A375 cells and HaCaT cells after 72 h of treatment. (**B**). Morphological changes of A375 cells and HaCaT cells with 6′-*O*-caffeoylarbutin, shown using an Olympus inverted biological microscope. * *p* < 0.05; ** *p* < 0.01.

**Table 1 ijms-25-00972-t001:** Main toxicity symptoms of 6′-*O*-caffeoylarbutin in different measurement groups (n = 20).

Group	Final Dose (mg·kg^−1^)	Toxic Symptoms (♀/♂)			
		Reduced Activities	Perianal Uncleanliness	Prone State	Shortness of Breath
Vehicle	-	0	0	0	0
CA ^a^ -1	28,056	8 (2/6)	0	2 (0/2)	4 (2/2)
CA-2	23,848	1 (0/1)	0	0	0
CA-3	20,272	0	0	0	0

^a^ CA denotes 6′-*O*-caffeoylarbutin.

## Data Availability

The data presented in this study are available on request from the corresponding authors.

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
