# Peer review of "6′-O-Caffeoylarbutin from Quezui Tea: A Highly Effective and Safe Tyrosinase Inhibitor"

_ijms, 2024, doi:10.3390/ijms25020972_

Round 1
Reviewer 1 Report
Comments and Suggestions for Authors
In the submitted manuscript Xie et. al. studied the potential mechanism of 6'-O-caffeoylarbutin as an inhibitor of mushroom tyrosinase (mTyr), involved in melanin biosynthesis. Various techniques, including high-performance liquid chromatography (HPLC), ultraviolet-visible (UV-Vis) spectroscopy, fluorescence spectroscopy, as well as molecular docking and molecular dynamics simulations were applied. Additionally, a safety assessment of 6'-O-caffeoylarbutin and preliminary research on its preservation efficacy in apple juice were conducted.
The manuscript is well-written and relevant to the field. In my opinion, the submitted manuscript could be accepted for publication in IJMS. However, there are a few comments that should be addressed before its publication:
Section Introduction
Line 70: The new figure with the structural formula of 6'-O-caffeoylarbutin would be beneficial for the readers.
Section Results
Figures 1-6: To improve the quality of all figures, the resolution and font size should be increased.
Subsection 2.3.2 MD simulation
Line 356: Molecular mechanism of 6'-O-caffeoylarbutin as an inhibitor of enzyme mTyr was studied through molecular docking and molecular dynamics simulations. In addition, molecular dynamics simulations coupled with free energy calculations could reveal the binding free energy of 6'-O-caffeoylarbutin at the active site of mTyr, which is in good agreement with experimental results.
References:
1. Pantiora, P. et. al. Monocarbonyl Curcumin Analogues as Potent Inhibitors against Human Glutathione Transferase P1-1. Antioxidants 2023, 12, 63. https://doi.org/10.3390/antiox12010063
2. Bren, U. et. al. Insight into Inhibitory Mechanism of PDE4D by Dietary Polyphenols Using Molecular Dynamics Simulations and Free Energy Calculations. Biomolecules 2021, 11, 479. https://doi.org/10.3390/biom11030479
Section Methods
The authors should provide the corresponding references to the applied protocols and methodologies in all subsections.
Lines 443-444: The authors extracted the 6'-O-caffeoylarbutin from Quezui Tea. However, the extraction protocol is not provided. The authors should include a new chapter on extraction methods.
Reference:
1. Bren, U. et. al. Helichrysum italicum: From Extraction, Distillation, and Encapsulation Techniques to Beneficial Health Effects. Foods 2023, 12, 802. https://doi.org/10.3390/foods12040802
Reviewer 2 Report
Comments and Suggestions for Authors
In this study, the authors used various methodological approaches to test and verify the inhibitory activity of 6'-O-caffeoylarbutin towards tyrosinase. I will mainly focus on the computational part of the study.
1. I suggest including the structure of 6'-O-caffeoylarbutin or, at least, arbutin in the Introduction to provide more information to readers.
2. Enlarge the size of Figures 2 and 3 for better visibility.
3. In the section on molecular docking, the authors state, "Based on the molecular docking results, it was found that 6'-O-caffeoylarbutin exhibits a favorable binding affinity with mTyr, with a docking score of -8.0 kcal∙mol-1, indicating a strong interaction between the compound and the receptor." This docking score should be compared to the score/binding affinity of a known tyrosinase inhibitor.
4. Please specify the atoms that participate in the hydrogen bonds observed in the complex obtained by molecular docking.
5. Figure 4 is not clear; the structures are tiny and hardly visible.
6. In Line 323: "Particularly within the range of 240-300 nm, the RMSF values of amino acid residues in the complex are significantly lower than those of mTyr protein." Please note that in the RMSF profile, the x-axis represents residues, i.e., their numbering in the primary amino acid sequence, not nanometers.
